# A Deep Diagnostic Framework Using Explainable Artificial Intelligence and Clustering

**DOI:** 10.3390/diagnostics13223413

**Published:** 2023-11-09

**Authors:** Håvard Horgen Thunold, Michael A. Riegler, Anis Yazidi, Hugo L. Hammer

**Affiliations:** 1Department of Compute Science, Faculty of Technology, Art and Design, Oslo Metropolitan University, 0176 Oslo, Norway; hhthunold@icloud.com (H.H.T.); michael@simula.no (M.A.R.); anisy@oslomet.no (A.Y.); 2Department of Holistic Systems, SimulaMet, 0176 Oslo, Norway

**Keywords:** clustering, deep learning, explainable artificial intelligence, image classification, knowledge discovery

## Abstract

An important part of diagnostics is to gain insight into properties that characterize a disease. Machine learning has been used for this purpose, for instance, to identify biomarkers in genomics. However, when patient data are presented as images, identifying properties that characterize a disease becomes far more challenging. A common strategy involves extracting features from the images and analyzing their occurrence in healthy versus pathological images. A limitation of this approach is that the ability to gain new insights into the disease from the data is constrained by the information in the extracted features. Typically, these features are manually extracted by humans, which further limits the potential for new insights. To overcome these limitations, in this paper, we propose a novel framework that provides insights into diseases without relying on handcrafted features or human intervention. Our framework is based on deep learning (DL), explainable artificial intelligence (XAI), and clustering. DL is employed to learn deep patterns, enabling efficient differentiation between healthy and pathological images. Explainable artificial intelligence (XAI) visualizes these patterns, and a novel “explanation-weighted” clustering technique is introduced to gain an overview of these patterns across multiple patients. We applied the method to images from the gastrointestinal tract. In addition to real healthy images and real images of polyps, some of the images had synthetic shapes added to represent other types of pathologies than polyps. The results show that our proposed method was capable of organizing the images based on the reasons they were diagnosed as pathological, achieving high cluster quality and a rand index close to or equal to one.

## 1. Introduction

Gaining new insights into diseases is crucial for diagnostics. Machine learning (ML) has been employed for this purpose. For instance, in genomics, ML has been used extensively to identify biomarkers for various diseases [1]. However, when patient data are presented as images, deriving insights into characteristics associated with a disease becomes more challenging. A common approach involves extracting a set of features from the images or performing image segmentation [2]. To search for new insights, these features can be analyzed using statistical or machine learning techniques. A recent advancement in this area is the use of graph convolutional networks on regions of interest from brain neuroimaging data [3,4]. Nevertheless, there is a risk that information might be lost during the feature extraction or image segmentation process. Another constraint is that the features and image segments typically rely on pre-existing medical knowledge, thereby limiting the potential for discovering new diagnostic insights.

To address these limitations, we introduce a novel framework based on deep learning (DL). The rationale behind this proposal is that the intricate patterns learned by DL models for successful predictions/classifications might also contain valuable insights into the relationship between the characteristics of medical images and the associated disease. The DL model operates independently of handcrafted features or human interventions, potentially overcoming the aforementioned constraints.

While there is great potential for DL to uncover new insights, this is still a highly unexplored area of research. This is likely because DL models are primarily designed for optimal prediction/classification performance, making them inherently difficult to interpret. Consequently, leveraging them to gain new medical insights can be challenging. However, the recent development of explainable artificial intelligence (XAI) provides methods to interpret ML and DL algorithms [5,6]. For instance, XAI techniques can highlight the sections of an input image that predominantly influenced the DL model’s decision, such as rendering a diagnosis. XAI is vital for quality-checking ML methods, ensuring they base their predictions on pertinent parts of the input. These methods also bolster trust in ML system recommendations among users.

In this paper, we propose a lesser-explored application of XAI, primarily to discover new knowledge. For example, XAI can be used to pinpoint features in an input image that clarify the DL model’s reasoning for classifying a patient as ill. A pertinent study in this context is the one by [7]. The authors trained DL algorithms to diagnose skin lesions, aiming to identify skin lesion biomarkers for various diagnoses. Their findings revealed that the “surrounding skin can also serve as evidence for skin lesion diagnosis,” which was previously omitted from traditional diagnostic procedures, signifying a potential new medical discovery. In [8], the authors harnessed XAI techniques to extract medical knowledge from electrocardiogram (ECG) data.

The diagnostic insights referred to above were discerned through a manual examination of explanations for multiple patients. Such a method, however, can be labor-intensive and subjective. Consequently, we introduce a pioneering framework that clusters multiple input patient images based on their explanations of why they were classified as sick or healthy. The practical implementation of this is intricate. We propose a unique technique that clusters the input images, giving more weight to the pivotal pixels based on the XAI explanation. This is accomplished using a Hadamard product between the input image and the explanation, producing what we term “explanation-weighted” images. Clusters derived from these images can provide an overview of varying explanations, some of which might elucidate novel disease characteristics. These clusters also help identify common explanations for a diagnosis (larger clusters) versus the less frequent, and perhaps less reliable, explanations (smaller clusters).

In summary, the framework proposed in this paper operates as follows: DL models are trained to distinguish patients with a disease from healthy controls or in another diagnostic context. XAI elucidates why patients were categorized as either ill or healthy. However, since XAI methods offer individual explanations for each patient, they cannot furnish an overview of various disease characteristics. Our proposed explanation-weighted clustering technique is designed to fill this gap. The main contributions of this paper are as follows:Introduction of a novel framework that integrates DL, XAI, and explanation-weighted clustering to unveil new insights into disease characteristicsComputation of so-called explanation-weighted images, which are obtained through the Hadamard product of an input image and its corresponding explanation.Development of a novel smoothing function to enhance the quality of XAI explanations.Evaluation of the framework on a large dataset comprising gastrointestinal tract images, consisting of real healthy images, real images with polyps, and images with a synthetic overlay to represent other disease types beyond polyps.

## 2. Background and Literature Review

We begin this section by providing an introduction and review of the techniques related to XAI, clustering, and cluster performance evaluation. Starting from Section 2.3, we offer an overview of methods that utilize ML to gain medical knowledge.

### 2.1. Explainable AI

XAI is a field of Artificial Intelligence (AI) that seeks to offer insights into black-box models and their predictions. Trust, performance, legal (regulation), and ethical considerations are some reasons researchers advocate for XAI [5]. This is increasingly critical as AI adoption reaches domains like healthcare.

External XAI techniques might explain single predictions through text or visualizations, or delve into models comprehensively using examples, local changes, or transformations to simpler models. While text and visualization explanations offer a direct, human-understandable clarification typically for a specific prediction, utilizing examples grants a broader understanding of a model by showcasing similar examples and predictions to the prediction in question. This method, however, does not provide an immediate explanation for a particular prediction. Local explanations focus on a subset of the problem, aiming to elucidate within that restricted context. Finally, to achieve higher interpretability, one can either employ a mimic model, which is an interpretable model that emulates the black-box model’s behavior, or replace the black-box model altogether. In this paper, we utilize visual explanations.

In [9], the authors detail four techniques that elucidate image classifiers by modifying the input. The methods include the method of concomitant variation, the method of agreement, the method of difference, and the method of adjustment. Each method offers a unique approach and insight into how models interpret and classify images.

The technical procedure to retrieve visual explanations from an image classifier comprises two parts: (1) an attribution algorithm furnishing the data for the explanation and (2) a visualization employing that data to generate a human-understandable elucidation. Broadly, image classification’s attribution algorithms can be classified as either gradient-based methods or occlusion-based methods.

Visualizations represent the interpretations derived from the attribution methods mentioned earlier. However, there is no consensus in the literature regarding what constitutes a “good” explanation. While some believe an explanation should detail parts of an image contributing to its classification, others focus on resolution quality or the trade-offs involved. Indeed, as 2D visualizations cannot fully depict a model’s intricacy, clarity about the limitations and trade-offs is essential when using such explanations.

Other research on improved visualization argues that past studies have overly concentrated on the positive alterations in an input image without contemplating the negative impacts [10,11]. Both perspectives are necessary for a comprehensive explanation, especially for AI adoption in sensitive areas.

### 2.2. Unsupervised Learning-Clustering

Clustering is an unsupervised ML technique where the objective is to discern groupings in unlabeled data. It has diverse applications, including anomaly detection, compression, or unveiling intriguing properties in data. In this paper, we focus on clustering for image classification using K-means and X-means clustering.

#### 2.2.1. K-Means and X-Means

K-means is a straightforward clustering algorithm with a time complexity of 
O(n)
 in big-O notation. The algorithm commences by initializing a centroid for each of the *K* clusters. Various strategies exist for this initialization. One method is to select K random points from the dataset as the initial centroids, although this can make K-means sensitive to its initialization. To mitigate this, one can execute K-means multiple times. Another method, K-means++, has been proven to be a more robust initialization scheme, outdoing standard K-means in both accuracy and time [12]. K-means++ selects initial centroids using a probability based on a point’s distance to the current centroids. Once initialized, each point is assigned to its closest centroid. The primary loop of the algorithm then adjusts the centroids toward the mean of the points linked to them, and points are reassigned to the nearest centroid.

A limitation of K-means is the necessity to predefine *K*, the number of clusters. If the number of clusters is unknown, one can employ X-means. This method involves running K-means algorithms with various *K* values. The most fitting number of clusters for a dataset can be determined by evaluating multiple clustering performance metrics.

#### 2.2.2. Clustering Performance Evaluation

Clustering performance evaluation metrics can be divided into two main categories: those requiring labeled data and those that do not. In this paper, we used the techniques Rand index, silhouette coefficient, and Davies–Bouldin index [13,14].

The Rand index measures the similarity between the labels that the cluster has assigned to data points and the ground truth labels. This metric differs from standard accuracy measures because, in clustering, the label of the cluster to which a data point is assigned may not match its true label. To accurately measure the performance of clustering, one must therefore account for permutations. The Rand index provides a score in the range [0, 1], indicating the number of matching pairs between the cluster labels and the ground truth. While it is highly interpretable, other methods must be employed when labels are not available.

The silhouette coefficient is a metric suitable for use when no labels are present. It produces a value that increases as clusters become more defined. “More defined” in this context means that the distance between points within a cluster is small, while the distance to other clusters is large. The silhouette coefficient yields a value in the range 
[−1,1]
: 
−1
 indicates an incorrect clustering, while 1 signifies highly dense clusters that are well separated.

In contrast, the Davies–Bouldin index places less emphasis on the quality of clusters and more on their separation. A low Davies–Bouldin index suggests a significant degree of separation between clusters. Its value starts at 0, which represents the best possible separation, and has no upper bound.

### 2.3. Image-to-Image Translation

Image-to-image (I2I) translation refers to the process of learning to map from one image to another [15]. Such a mapping could, for instance, transform a healthy image into one with pathological identifiers. Differences between the input and output images can then be analyzed to extract medical insights. For this purpose, generative adversarial networks (GANs) and variational autoencoders (VAEs) have been employed.

RegGAN has proven to be the most effective I2I solution for medical data [16]. One challenge of I2I in the medical realm is the difficulty in finding aligned image pairs in real-world scenarios. To address this, the authors used magnetic resonance images of the brain and augmented them with varying levels of noise and synthetic misalignment through scaling and rotation. RegGAN surpassed previous state-of-the-art solutions for both aligned and unaligned pairs and across noise levels ranging from none to heavy.

In the realm of I2I translation, there are also initiatives leveraging newer architectures, such as Transformers. Specifically, the Swin transformer-based GAN has demonstrated promising results on medical data, even outperforming RegGAN on identical datasets [17].

### 2.4. Data Mining Techniques

Data mining involves extracting valuable knowledge from large datasets by identifying pertinent patterns and relationships [18]. Clustering is one such technique and is also employed in this paper. While clustering has found application in diverse facets of medical knowledge discovery, its direct use in medical imaging remains relatively rare.

In [19], the authors demonstrated that K-means clustering could identify subgroups of yet-to-be-treated patients. This approach unveiled four unique subgroups. The results were promising for K-means clustering, but unlike our study, their research did not incorporate images.

In another study [20], a system was proposed for medical image retrieval. The process began by searching for an image within a known primary class and subsequently by the identified markers that had not been labeled previously. The authors showcased that by using clustering, previously unlabeled subclasses could be detected, facilitating the search for analogous images. This proved beneficial, enhancing a doctor’s diagnostic accuracy from 30% to 63%.

Other research focuses on how data mining techniques can offer insights not directly as new medical knowledge from AI, but rather to equip users with enriched information, allowing them to derive novel medical insights. In [21], a visualization solution was proposed for practitioners, grounded in spectral clustering, to decipher information from 2D and 3D medical data. Although spectral clustering was central to their approach, they recognized that no single clustering method excels universally.

### 2.5. Explainable Artificial Intelligence

In the medical domain, XAI research predominantly serves ethical and legal imperatives, fostering trust and privacy, and revealing model biases [22]. The deployment of XAI for medical knowledge discovery is less common, yet some recognize its untapped potential [6].

In [23], the authors illustrated a method to cluster images, assign groups of images importance scores, and subsequently obtain explanations regarding significant components across an entire class. This technique employs super-pixel segmentation to fragment the image. This procedure is replicated for all images in a class. The segments are then clustered, and the significance of each segment group is evaluated. This approach yields explanations highlighting crucial features across the class. Although their evaluation used a general dataset, it appears feasible to adapt this to the medical context. In such cases, this methodology could potentially expose medical insights by categorizing types of markers. This aligns with the objectives of this paper, albeit through a different modality. Here, the explanations are the primary outcome, contrasting with our work where explanations are integral to the process of enhancing the categorization.

In [24], the authors exemplified how XAI can be harnessed for medical knowledge discovery. Using a ResNet, they autonomously analyzed ECG data. Their model could predict the sex of a subject with 86% accuracy, a feat they claim is nearly unachievable for human cardiologists. To elucidate the model’s learned insights, they turned to XAI. They modified Grad-CAM, presenting a visual justification of the segments deemed crucial for the prediction. This process revealed what the authors termed as “new insights into electrophysiology”. While their study did not incorporate images as in our research, it did utilize 2D visual explanations to uncover fresh medical knowledge.

No papers were identified that directly leverage the explanations of image classifiers in a manner analogous to the methodology proposed in this paper.

## 3. Methodology

In this section, we present our proposed novel framework that combines DL, XAI, and clustering to potentially uncover new insights into characteristics in medical images associated with a disease.

We recognize that individuals can be diagnosed with a disease based on different criteria. For example, a doctor might determine a mole’s cancer risk based on various factors such as color, spots, irregular border, or, as identified by [7], the surrounding skin. An overview of the method we developed and implemented in this paper is shown in Figure 1. The green border represents healthy data, while the red with a dotted border indicates the flow of pathological data. In this example, we envision two characteristics of the disease, represented by blue and yellow shapes. Our framework aims to automatically identify these two characteristics. It is important to note that both fall under the “sick” class, and since we do not have labeled data to distinguish them, traditional machine learning cannot be employed. Instead, we propose using XAI and clustering to achieve this distinction.

Our proposed procedure includes the following steps:Step 1We train a DL image classifier on healthy and pathological data. We detail the data in Section 3.1 and Section 3.2. Section 3.3 describes the DL classifier we chose. As shown in Figure 1, the pathological class consists of two characteristics, namely the yellow and blue shapes.Step 2We predict a set of images that the DL method has not seen and compute explanations for each image classified as pathological, using XAI techniques. We elaborate on this step in Section 3.4.Step 3We create an explanation-weighted version of the input images by computing the Hadamard product between the input image and the visual explanation.Step 4We cluster the explanation-weighted images from Step 3 to identify the different diagnostic characteristics in the pathological class, illustrated in the figure as yellow and blue shapes. We detail the clustering methodology in Section 3.5.

One might question why, in our proposed framework, we chose to cluster the explanation-weighted images classified as pathological rather than the original images. The rationale is that clustering the original images, classified as pathological, offers no assurance that they would be clustered based on pathological patterns. They could be grouped according to other unrelated image attributes, like brightness levels. In our experiments (though not included in the paper), we found that without the explanation and weighting provided in steps 2 and 3, the images were clustered based on criteria other than the reasons they were classified as pathological.

The methodology can be further equipped with uncertainty measures, giving us the ability to identify which characteristics (clusters) are statistically significant findings and which are just random noise. However, we have not explored uncertainty measures in this paper.

### 3.1. Gastrointestinal Dataset

The dataset we used as the basis for all experiments in this work is the HyperKvasir dataset [25]. It is a large image dataset of the GI tract taken from gastro- and colonoscopy examinations performed at Bærum Hospital in Norway. The images were captured using a Pentax colonoscope (Pentax Medical Europe, Hamburg, Germany). Some images contain extra information in the form of a picture-in-picture located in the bottom left corner, recognizable by its distinct green background. These images were taken by an Olympus ScopeGuide™, a device used to image the colon (Olympus Europe, Hamburg, Germany).

The dataset comprises 10,662 labeled and 99,417 unlabeled images. Each label was reviewed by more than one expert in the field and is, therefore, assumed to be highly accurate. The dataset also includes videos containing 889,372 video frames. Figure 2 displays a sample from the dataset.

### 3.2. Pseudo-Real Data

We created two pseudo-real datasets from the HyperKvasir dataset for the experimental parts of the paper, referred to as Datasets A and B. By `pseudo-real’ dataset, we refer to a dataset consisting of real medical images and real medical images with a small colored shape added to represent pathology.

Dataset A comprises 20,000 samples; 10,000 of these were real images from the HyperKvasir dataset representing the healthy class. The other 10,000 samples were created using the same images, but colored shapes were added to each image: either a single yellow rectangle or a blue ellipse. The width and height of these shapes were randomly set between 20–25% of the images, with each shape added at a rate of 50%. Examples of healthy and pathological samples are shown in Figure 3.

Dataset B contains 2056 images labeled as healthy in the HyperKvasir dataset and 2056 images, with half being real pathological images, in the form of polyps, and half being pseudo-real, in the form of blue shapes. We chose the blue ellipse because it resembles polyps in terms of size and shape, adding to the dataset’s challenge. Examples of healthy and pathological samples are shown in Figure 4. We see that half of the images in the pathological class consist of real polyps and half-blue shapes.

The HyperKvasir images have an aspect ratio of 4:5. The classifiers we used in this paper required square images as input. Therefore, we cropped the images to a 1:1 aspect ratio. A bottom-left crop ensured that the entire Olympus ScopeGuide image was included. This process does remove 20% from the top of the images, resulting in some information loss. However, this approach aligns with the methods used in the official experiments by the HyperKvasir dataset creators. We further scaled the images to 
224×224
 pixels, as expected by the classifiers used in the experiments in this paper, and normalized them between 0 and 1. Finally, we augmented the training data with a 50% chance of randomly flipping the image horizontally or vertically and rotating it uniformly up to 90 degrees.

Setting up the pseudo-real datasets in this manner serves three main purposes. First, by using real data for the healthy class and the same images as the background for the pathological class, the classification problem becomes more realistic and challenging. There is a potential for the classifier or the XAI method to confuse the actual difference between the classes (the colored shapes) with other image characteristics, such as polyps or the green shapes in the bottom left corner of some images [25]. Second, since we know the true explanation (given by the presence or absence of the colored shapes for Dataset A and colored shapes and polyps for Dataset B), it allows us to evaluate the performance of different XAI explanation methods. Third, with the number of possible synthetic classes only being limited by the combinations of colors and shapes, the pseudo-real dataset can be constructed to be as simple or complex as needed.

### 3.3. Classification

We selected the ResNet architecture with 50 layers (ResNet50) for the experiments conducted in this work [26]. Table 1 provides details of the architecture. Furthermore, we followed the ResNet50 model architecture up to the classification head. This classification head is task-specific and is replaced with a single neuron, which provides a prediction in the form 
$\hat{y}\in R|0\leq \hat{y}\leq 1$
 after the softmax is applied. We chose this architecture because it has demonstrated good results in classification tasks within medical imaging. Additionally, it is a model that previous research on XAI has employed to extract explanations, making it a reliable base model for our experiments. The method we propose in this paper does not necessitate a specific architecture. We should also note that the main aim of this paper is not to identify the best possible DL model for the classification problem. Instead, we aim to select a model that performs well to demonstrate the efficacy of the suggested framework, as described at the beginning of Section 3.

### 3.4. Explanations

We considered a multitude of methods for extracting explanations from image classifiers, each with its benefits and trade-offs. However, we selected the occlusion-based method by [27] as it consistently performed the best in our experiments. Since we had synthetic overlays and annotation (polyp), we were privy to the true explanation, which we could then use to compare the performance of various explanation methods.

Given an image *I* with width and height 
W×H
 and three color channels, 
$I\in R^{W\times H\times 3}$
, and a black-box classifier 
$f\left(I\right)\to \hat{y}\in R|0\leq \hat{y}\leq 1$
, we obtain an explanation by modifying *I* and recording the changes in 
$\hat{y}$
 for each pixel *j*. The modification involves a 2D patch of pixels *P* of size 
$P_{W}\times P_{H}$
 moved over the image with stride *s*. We apply the patch *N* times until all pixels are covered. Each patch is added to a copy of the original image, and the colors of the affected pixels are replaced with gray color values (128 in all color channels for RGB 0-255). As the stride increases, the computation decreases, but it results in reduced detail. We chose gray because most of the research we reviewed used this color. Let us represent this color by the constant *C*. For easier computation, the patch *P* can match the image’s first two dimensions, with 0’s in the patch position and 1’s elsewhere, facilitating clean matrix multiplication. The occluded input using a given patch can then be expressed as

(1)
g(I,P)=f(I⊙P+(1−P)⊙C)

where ⊙ denotes the Hadamard product. The importance of some pixel 
j∈I
 can then be calculated as the average change in classification probability across all patches containing *j*. Specifically, let 
$P_{i},i=1,\ldots ,N$
 denote all the patches used, and let *J* represent all patches containing pixel *j*, i.e., 
$J=\{i:j\in P_{i},i=1,\ldots ,N$
}. Then, the importance of pixel *j*, denoted 
$Y_{j}$
, is computed as

(2)
$$Y_{j}=\frac{1}{\left|J\right|}\sum_{j\in J}\left(f\left(I\right)-g\left(I,P_{j}\right)\right)$$

where 
|J|
 indicates the number of patches containing *j*. Also, let *Y* symbolize the importance of all pixels in image *I*.

For improved visualization, we typically normalize *Y* values between 0 and 1 for all positive values and between 0 and −1 for all negative values. Although the attribution methods retain only the positive values, research on visualizations indicates that negative values play a crucial role in forming comprehensive explanations, prompting us to include them in our method. We have omitted the mathematical formulation of the normalization procedure for the sake of brevity.

Moreover, we propose applying a smoothing function, *S*, to the importance map to eliminate noise from less significant pixels and accentuate those of greater importance. This step drew inspiration from [28], where values are clipped, setting a clear boundary between important and unimportant pixels. However, our experiments showed that a smoothing function produced superior visualizations. This function takes three parameters: the value to smooth, *x*, the offset 
θ
, and the strength 
σ
:
(3)
$$S\left(x,\theta ,\sigma \right)=\left\{\begin{array}{cc} \frac{x^{\sigma }}{\theta^{\sigma -1}}, & \mathrm{if}\;x\leq \theta \\ 1-\frac{{\left(1-x\right)}^{\sigma }}{{\left(1-\theta \right)}^{\sigma -1}}, & \mathrm{otherwise} \end{array}\right.$$


The final explanation is then defined as 
S(Y,θ,σ)
, where 
θ
 and 
σ
 are hyperparameters that must be chosen for each use case. The 
θ
 hyperparameter adjusts the function’s point where values transition from being suppressed to being amplified. A higher 
θ
 results in more noise removal. Conversely, the second hyperparameter, 
σ
, determines the function’s rate of change. We illustrate the smoothing function using 
θ=0.1
 and 
σ=8
 in Figure 5. In a manner similar to [28], we have not identified an automatic procedure to pinpoint optimal hyperparameter values and have, therefore, tuned the parameters based on visual inspection. The resulting smoothing function was robust in our experiments, consistently performing well for all images. The function solely enhances the explanations’ clarity, and we did not detect any bias introduction in the explanations due to it.

As elaborated above, this method solely depends on modifying the model’s input and tracking output changes, making it model-agnostic. This characteristic is vital for the method’s future applications, enabling the selection of the best model based on its proficiency in identifying characteristics in medical imaging, rather than its compatibility with the XAI method. This flexibility contrasts with the rigid coupling that certain gradient-based methods might possess with the model architecture. Another compelling reason for our choice of an occlusion-based approach is its documented ease of comprehension for humans [29], a critical factor in the medical realm where professionals without ML expertise must interpret an explanation.

### 3.5. Explanation-Weighted Clustering

From the first three steps of the procedure, listed at the beginning of Section 3, several images are classified as pathological, and we provide an explanation in terms of pixel importance *Y* for each image. Recall that we consider the situation where we imagine that the images are classified by the DL model due to different characteristics, identified by the occlusion-based method.

In this section, we describe the fourth step of the proposed procedure. Here, we employ clustering to gain an overview of the characteristics pinpointed by the occlusion-based method. We achieve this by emphasizing the crucial pixels in terms of the explanations *Y*. Essentially, we group the images based on different types of explanations or pathologies.

For the pathological class, we are keen on understanding the pixel’s importance in relation to pathology. Therefore, we set all values of *Y* less than zero to zero, denoting it as 
$Y_{+}$
. We compute the explanation-weighted images, referred to as 
$I_{Y}$
, by performing a Hadamard product between the input image, *I*, and the non-negative explanation 
$I_{Y}=I⊙Y_{+}$
.

Direct clustering of the explanation-weighted images is not feasible when dealing with 50 thousand features (pixels) for a 
224×224
 image, and becomes even more challenging for higher resolution images, like 
1024×1024
. Thus, we needed a feature extraction mechanism. We achieved this by running the explanation-weighted images through an image classifier and then using the output, just before the classification layer, as features for clustering. As detailed in Section 3.3, the penultimate layer is the average pooling layer, which outputs a 256-dimensional vector. While this vector could be clustered directly, its 256 dimensions are not optimal concerning computational efficiency and clustering accuracy. Hence, we reduced it to a lower dimension using Singular Value Decomposition before clustering [30]. We employed K-means for clustering. However, since K-means necessitates specifying the number of clusters, *K*, it does not align seamlessly with our method, particularly when the number of explanation clusters (different pathologies) is typically expected to be unknown. Thus, we determined the number of clusters by clustering repeatedly with an increasing number of clusters, subsequently evaluating the cluster metrics. We selected the *K* with the highest silhouette coefficient. We chose the silhouette coefficient as our metric since it provides a tangible number showcasing how distinct the clusters are. The rationale behind this is that well-defined clusters (with a high silhouette coefficient) suggest correct data clustering since segregating a well-defined cluster is challenging. In contrast, poorly defined clusters (with a low silhouette coefficient) hint at the possible existence of multiple clusters within a given cluster. While this approach can yield more than one plausible solution based on the data, the same holds true for other methods to determine *K*, such as the elbow method. In the elbow method, the inertia is plotted, and the graph’s elbow point signifies the correct cluster count.

## 4. Experiments and Results

To develop efficient classifiers for Datasets A and B as described in Section 3.2, a pre-trained version of ResNet50 on 1000 classes of ImageNet [31], was fine-tuned for Datasets A and B using transfer learning. The datasets were split into training, validation, and test sets with 64%, 16%, and 20% of the data in each set, respectively. A batch size of 64, the binary cross-entropy loss function, and stochastic gradient descent with a learning rate of 0.001 and a momentum of 0.9 were used. The learning rate was decayed by a multiplicative factor of 0.1 every seven epochs. The model was trained with these parameters until early stopping was triggered, which was based on no improvement in the F1 score on the validation set for five consecutive epochs. The models were trained on a Nvidia V100 GPU.

In the occlusion XAI method on Dataset A, a patch size of 
$P_{W},P_{H}=24$
 and a stride 
s=8
 were used, and on Dataset B, a patch size of 
$P_{W},P_{H}=64$
 with a stride of 
s=16
 was used. For both datasets, 
θ=0.1
 and 
σ=8
 were used in the smoothing function, as detailed in Equation (Equation 3).

Throughout the rest of this section, we present the results from the experiments detailed above. The fine-tuned ResNet50 classifiers learned to classify the test examples of both Dataset A and B with perfect test scores, i.e., test examples were classified correctly. This was even better than expected, especially for Dataset B, where 75% of the test examples were real data without any synthetic overlay, i.e., all the real healthy images must be classified as healthy and all the real polyp images (half of the pathological class) must be classified as pathological. Classifying polyps from healthy data is known to not be a straightforward task [32]. Next, we analyze the performance of the quality of the explanations and the explanation-weighted clustering.

### 4.1. Explanations

Figure 6 displays two images. The image on the left presents a pseudo-real sample with a blue ellipse symbolizing a pathological identifier. In contrast, the image on the right overlays the explanation onto the image. Green regions indicate areas positively influencing the model’s prediction of the pathological class, whereas red regions denote a negative impact. Greater color brightness and visibility signify higher importance. The explanation underscores a pronounced focus on the anticipated area (the blue ellipse). This indicates that the occlusion method has accurately discerned the identifiers of the pathological class. This observation holds consistent upon visually analyzing a broader set of samples. The insights from the explanations suggest that the classifier does not exhibit any evident bias.

Figure 7 displays an image of a polyp (the small growth in the upper right corner) both with and without the overlaid explanation. The explanation highlights the most pronounced and expansive focus on the upper portion of the polyp, whereas the base of the polyp receives minimal attribution for the classification. Unlike the explanation for the pseudo-real image discussed earlier, this image identifies attribution in multiple areas. These areas, being less bright, suggest diminished importance. Whether these areas are medically significant cannot be determined without the input of a medical professional.

While the focus is appropriately directed, the samples in Figure 6 and Figure 7 reveal that it is slightly offset towards the top left of the pathological identifier. We believe this is likely a result of the combination of the chosen attribution method, which applies gray rectangles from the top-left to bottom-right, the selected patch size, and the classifier architecture. This conclusion is drawn from the observation that this offset is consistently present across all classes and in explanations from both Datasets A and B. Despite these weaknesses with the occlusion-based method, it performed better in our experiments than gradient-based methods. Delving deeper into the source of the offset would necessitate further research, but is beyond the focus of this paper. The focus of the paper is to demonstrate the usefulness of the suggested medical knowledge discovery framework, rather than improving existing DL, XAI, and clustering methods. The alignment with the anticipated focus and the level of detail captured also hinge on the chosen hyperparameters. Figure 8 illustrates the relationship between patch size and explanation accuracy. The top row of images presents the image with a gray patch for size reference, while the bottom row displays the corresponding explanations. Beginning on the left, the figure showcases patch sizes of 64 pixels, followed by 48, 32, and concluding with 24 pixels. The first patch, approximately 150% the size of the ellipse, results in significant attribution to areas outside the ellipse. The subsequent patch, roughly equivalent in size to the ellipse, still attributes large areas outside of the ellipse, albeit to a lesser degree than the preceding patch. Shrinking the patch to 75% of the ellipse’s size yields a more precise attribution area. Further diminishing the patch size to 50% of the ellipse’s size reduces the extraneous attribution but also omits some accurate attribution from the ellipse. We observe that the explanations are profoundly influenced by the chosen patch size. As patch size enlarges, recall surges, and as it diminishes, precision intensifies. This presents a balancing act we must consider.

### 4.2. Explanation-Weighted Clustering Experiments

The images on the left side of Figure 9 display the original pathological images from Dataset A. In contrast, the images on the right present the explanation-weighted images, which result from applying the Hadamard product to the images on the left and the explanations, as detailed in Section 3.5. We observe that the resulting explanation-weighted images effectively isolate the pathological characteristics.

Table 2 displays the clustering performance using two distinct sets of features. The first row (pre-trained) pertains to features derived from the pre-trained ResNet50 on ImageNet, while the second row (fine-tuned) pertains to features from the ResNet50 after fine-tuning it to distinguish between healthy and pathological images in Dataset A. We observe that both sets of features deliver commendable performance, with a rand index close to, and at times equal to, one. This indicates that one cluster comprises the yellow shapes while the other encompasses the blue shapes. The features from the fine-tuned model exhibit a slight edge in terms of cluster quality.

The images on the left side of Figure 10 display samples from the pathological class of Dataset B. The images on the right present the results after explanation weighting. Once again, we observe that the method efficiently isolates the pathological characteristics. However, the precision is slightly lower than for Dataset A. This outcome aligns with our expectations, given that classifying Dataset B is more challenging than Dataset A. This complexity arises because 50% of the pathological class in Dataset B comprises real pathologies (polyps).

Table 3 displays the quality of the clusters for Dataset B. Using the pre-trained features yields a rand index of 0.92, while employing the fine-tuned features for this dataset produces a perfect rand index of one. We observe that the cluster quality improves when using the fine-tuned features. By comparing Table 2 and Table 3, as anticipated, we notice that the cluster quality is superior for Dataset A compared to Dataset B. Overall, our results remained robust across different variants of the datasets and initial values of the clustering algorithm.

## 5. Discussion

In this paper, we developed a framework that leverages DL, XAI, and clustering to potentially uncover new characteristics associated with a disease. Offering credible explanations for DL methods is notoriously challenging, as these explanations can be marred by noise and bias. Hence, in this paper, we advocate for clustering explanation-weighted images. This approach enables us to gain an overview of distinct groups of explanations or pathological patterns. Some of these clusters could discover new insights, with the size of a cluster indicating the prevalence of a particular characteristic.

### 5.1. Faithfulness vs. Interpretability

A pivotal element of the method we propose in this paper is the visual explanation. Within the realm of visual explanations, it is imperative to consider both faithfulness and interpretability. Faithfulness in visual explanations refers to the model’s ability to authentically represent the learned function. It is important to recognize that a balance between faithfulness and interpretability is often necessary. While faithfulness is pivotal, ensuring that explanations are lucid and easily comprehensible is crucial for effective communication. Raw explanations can sometimes be hard to interpret. Often, some processing is employed to refine the explanation and enhance its interpretability. In [28], the authors contend that human perception of colors is not linear. Consequently, amplifying values to make them more discernible is deemed reasonable; otherwise, one might overlook the significance of a particular image area. In their study, pixels with high attribution were clipped. This approach resulted in a minimal gradient difference among high-end pixels. To ensure a better gradient distinction between attributed pixels, our work employs a smoothing function that amplifies values based on a designated function. A hyperparameter can then govern the extent of this amplification, influencing the faithfulness of the visualization. Achieving the right equilibrium between faithfulness and interpretability is vital for crafting visual explanations that are both truthful and easily understood. In this work, we introduced a unique smoothing function to the explanation. This function curbs noise and accentuates the prominence of crucial pixels. This approach not only preserves the faithfulness of the learned function representation but also enhances interpretation. Such adjustments enhance clustering efficiency and aid in the clearer comprehension of the resulting clusters.

### 5.2. Strengths and Weaknesses

Given that utilizing DL and XAI for knowledge discovery remains a largely untapped area of research, the novelty of our study stands out as a strength. Methodologically speaking, the employment of pseudo-real data is worthy of mention. Using pseudo-real data makes the findings more palatable to a broader AI audience since medical expertise is not a prerequisite for comprehension. This approach ensures that our work can be accurately evaluated and generalized across various domains. Nevertheless, there is an associated drawback. Pseudo-real data do not fully encapsulate the efficacy of the proposed methodology when applied to genuine medical images. Medical professionals were not involved in this research, limiting the evaluation scope on authentic medical data. Nonetheless, the results from Dataset B, which focused on real-life polyps, suggest promising prospects for real-world medical applications.

Any potential medical insight discovered using the suggested framework must be verified by follow-up clinical trials before being implemented in clinical practice. This is because approaches using machine learning for knowledge discovery do not satisfy the requirements of the hypothetico-deductive model. However, machine learning-based methods, including the suggested framework in this paper, can be highly useful for developing new and falsifiable medical hypotheses that can be further tested in clinical trials. Developing medical hypotheses is a fundamental part of advancing the field of medicine.

## 6. Conclusions

Gaining new medical knowledge from image data is an exceedingly challenging task. A prevalent approach in the literature involves manually extracting features from images and then analyzing the occurrence of these features in images associated with a disease in comparison to healthy controls. However, a challenge with this approach is the potential loss of crucial information during the feature extraction process. Additionally, these features are often derived based on pre-existing medical knowledge, which can constrain the discovery of novel insights.

In this paper, we proposed a novel framework that combines DL, XAI, and clustering to uncover new insights from medical image data. This framework sidesteps the need for manual feature extraction and human intervention, offering a potential solution to the limitations inherent in feature extraction-centric approaches.

### Future Work

The results presented in this paper underscore the substantial promise of our novel framework. Nonetheless, a limitation of our analyses is the presence of synthetic overlays in some of the images from Datasets A and B. A logical progression would be to apply our method to real data devoid of synthetic components. However, this endeavor would necessitate meticulous planning and collaboration with clinical experts, which extends beyond the scope of a single research paper.

In principle, our proposed framework could be applied across various domains, including economics, biology, climate science, and social studies. For instance, there is significant interest in climate research to comprehend weather patterns leading to extreme weather events [33]. Our proposed technique could potentially be harnessed to analyze distinct explanations or weather states that precipitate future extreme weather events. Delving into the applicability of our framework in other disciplines presents an exciting avenue for future investigations.

Generating visual explanations for DL models is generally a complex endeavor. A possible enhancement could involve training multiple DL models, employing various XAI techniques, and devising strategies to determine a consensus across these models and XAI outputs. This consensus could subsequently serve as the foundation for explanation-weighted clustering.

Moreover, it might be worthwhile to examine the scalability of our framework, especially when handling larger datasets or more intricate multi-modal medical data scenarios.

To further our exploration, conducting additional experiments focused on the role of DL in clustering could be enlightening. Probing into explanation-weighted clustering using cutting-edge deep clustering techniques could offer significant advancements and innovations.

## Figures and Tables

**Figure 1 diagnostics-13-03413-f001:**
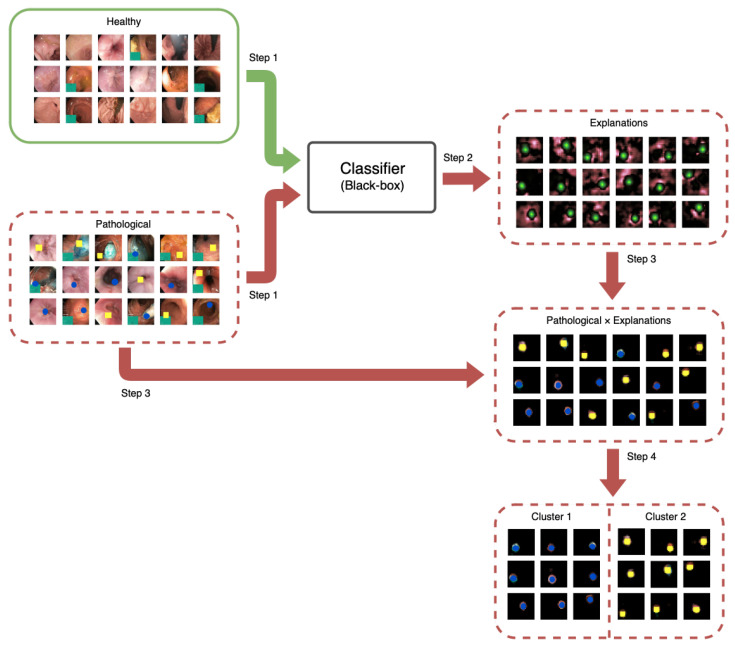
Overview showing the four main steps of our proposed framework.

**Figure 2 diagnostics-13-03413-f002:**
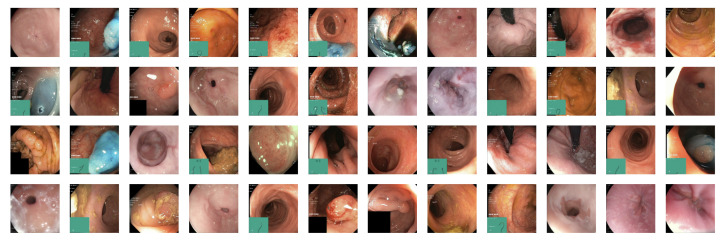
A sample from the HyperKvasir dataset showing images from the upper and lower GI tract.

**Figure 3 diagnostics-13-03413-f003:**
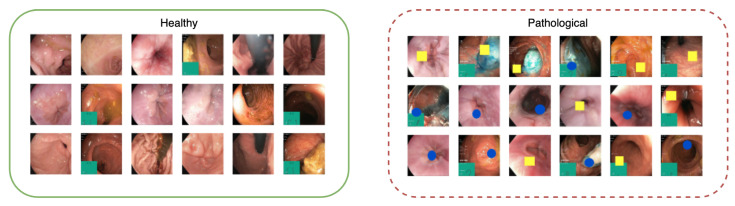
Examples of images from Dataset A with images from the healthy class on the (**left**) and the pathological class on the (**right**). The pathological class consists of the same images as the healthy class but with blue or yellow synthetic overlays representing two types of pathologies.

**Figure 4 diagnostics-13-03413-f004:**
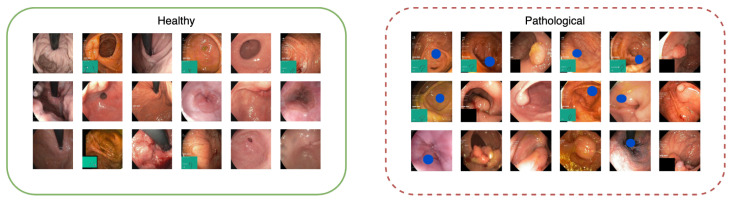
Examples of images from Dataset B with images from the healthy class on the (**left**) and the pathological class on the (**right**). The pathological class features two pathologies: real polyps and blue shapes.

**Figure 5 diagnostics-13-03413-f005:**
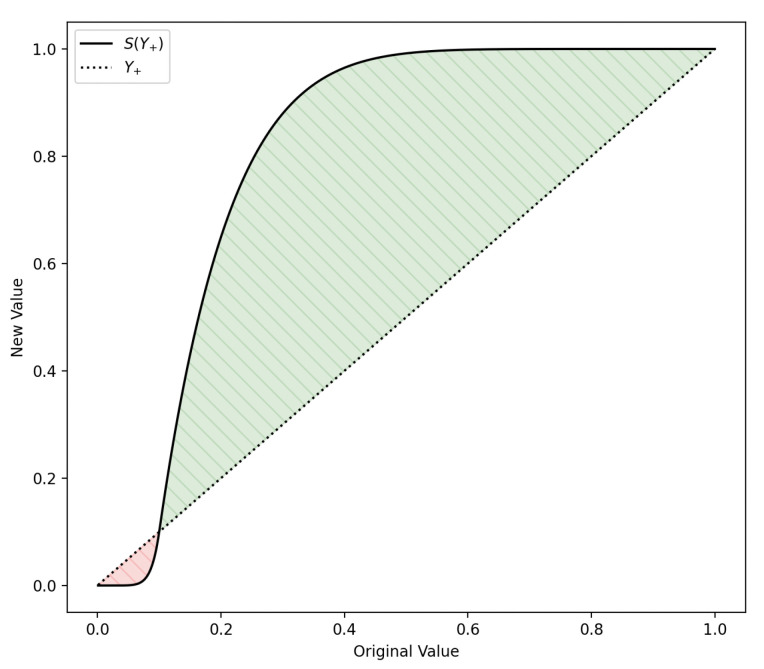
Visualization of the smoothing function in Equation (Equation 3) using 
θ=0.1
 and 
σ=8
 applied to the explanations.

**Figure 6 diagnostics-13-03413-f006:**
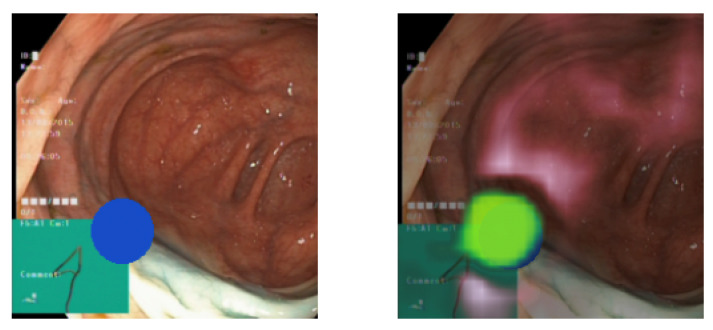
(**Left**) A pseudo-real data sample. (**Right**) The sample overlaid with its explanation.

**Figure 7 diagnostics-13-03413-f007:**
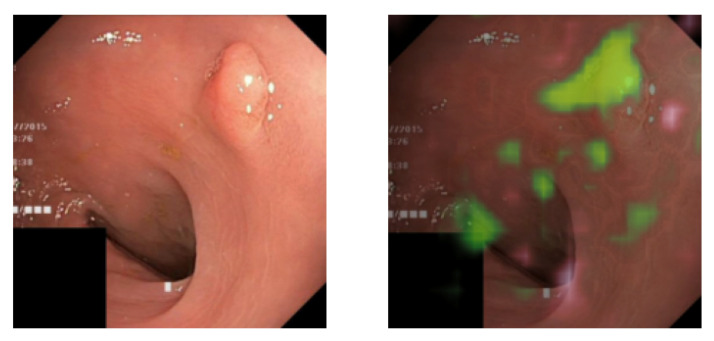
(**Left**) A real data sample of a polyp. (**Right**) The sample overlaid with its explanation.

**Figure 8 diagnostics-13-03413-f008:**
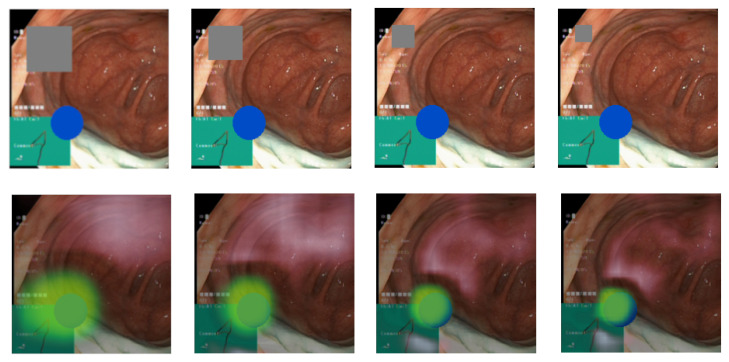
Visualization of how the patch size affects the explanation. (**Top row**) The gray square shows the patch size and the blue ellipses the pathology. (**Bottom row**) The green overlay shows the resulting explanation using the patch size in the image above.

**Figure 9 diagnostics-13-03413-f009:**
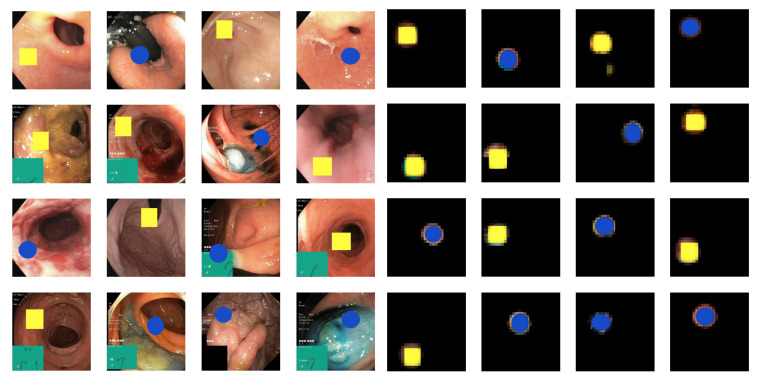
Original (**left**) and explanation-weighted (**right**) image samples from the pathological class of Dataset A.

**Figure 10 diagnostics-13-03413-f010:**
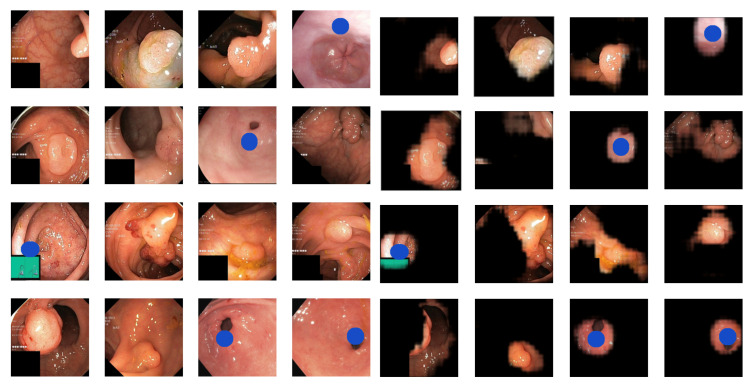
Original (**left**) and explanation-weighted (**right**) image samples from the pathological class of Dataset B. The pathologies in Dataset B are real polyps and blue ellipses.

**Table 1 diagnostics-13-03413-t001:** Classifier architecture overview.

Input $x\in R^{224\times 224\times 3}$
Conv 64, 7 × 7, stride = 2, BN, ReLU
Max Pool, 3 × 3, stride = 2
Residual Block × 16
Average Pool, 2 × 2, stride = 2
Fully-connected 1
Output $\hat{y}\in R|0\leq \hat{y}\leq 1$

**Table 2 diagnostics-13-03413-t002:** Clustering results for Dataset A. The first column represents the features used, the second column the results of a classification test using images previously not seen by the cluster and the third column shows cluster quality metrics. Bold indicates the best values.

Features	Classification (Rand Index)	Cluster Quality (Silhouette/Davies–Bouldin)
Pre-trained	0.995	0.485/0.840
Fine-tuned	**1.0**	**0.548**/**0.724**

**Table 3 diagnostics-13-03413-t003:** Clustering results for Dataset B. The first column represents the features used, the second column the results of a classification test using images previously not seen by the cluster and the third column shows cluster quality metrics. Bold indicates the best values.

Features	Classification (Rand Index)	Cluster Quality (Silhouette/Davies–Bouldin)
Pre-trained	0.919	0.259/1.567
Fine-tuned	**1.0**	**0.431**/**0.979**

## Data Availability

The data used in this research article are open-access and can be downloaded via https://datasets.simula.no/hyper-kvasir/ (accessed 7 November 2023).

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
