# Peer review of "A Deep Diagnostic Framework Using Explainable Artificial Intelligence and Clustering"

_diagnostics, 2023, doi:10.3390/diagnostics13223413_

Round 1

Reviewer 1 Report

Comments and Suggestions for Authors

This manuscript presents a novel framework for extracting actionable knowledge from deep learning (DL) models trained on medical imagery, utilizing Explainable Artificial Intelligence (XAI) and "explanation-weighted" clustering techniques. The authors have applied their methodology to gastrointestinal tract images, where synthetic shapes were utilized to represent different pathological conditions. The proposed framework aims to generate a better understanding of the model’s decision-making process, and potentially unveil new features characterizing diseases, thus demonstrating an intriguing interface between XAI, DL, and medical image analysis.

1.       The manuscript could be strengthened by including a more robust comparison with existing methodologies in the realm of XAI and DL in medical imagery.

2.       The lack of involvement of medical professionals in the validation of the results, especially on real medical data, is a significant limitation. The clinical validity of the generated explanations and discovered knowledge could greatly enhance the impact and applicability of the proposed method.

3.       The experimentation mostly hinges on synthetic data and synthetic pathology, which may not fully reflect the complexities and variations encountered in real pathological conditions.

4.       Some sections, like the discussion on faithfulness versus interpretability and the specific trade-offs made in the smoothing function for better visualization, are insightful but could be expanded for a better understanding of the implications.

5.       The offset observed in the explanation visualization is acknowledged but not thoroughly investigated. A deeper analysis of this offset and its potential impact on the explanation quality is warranted.

6.       It would be beneficial to have a more exhaustive comparative analysis with existing methodologies, highlighting the unique advantages and possible limitations of the proposed framework.

7.       Engaging medical professionals for a more clinically grounded validation of the generated explanations and discovered knowledge is highly recommended.

8.       A more detailed analysis of the observed offset in explanation visualizations, its implications, and possible solutions should be explored.

9.       Extending the experimentation to more varied and complex real-world medical datasets, and perhaps, incorporating a multi-class scenario could provide a more rigorous validation of the proposed method.

10.   The potential impact of the chosen hyperparameters on the explanation and clustering results could be discussed more thoroughly, possibly through a sensitivity analysis.

11.   In future work, it might be useful to explore the scalability of the proposed framework to larger datasets and more complex, multi-modal medical data scenarios.

Overall, this manuscript offers a valuable contribution to the field of XAI in medical image analysis, providing a novel methodological framework for knowledge discovery. However, further validation on real medical data, in collaboration with medical professionals, and a more robust comparison with existing methods could significantly enhance the impact and the applicability of the proposed method.

Comments on the Quality of English Language

 Extensive editing of English language required

Author Response

Dear reviewers

Thank you for your valuable comments to our paper. Please find attached our response letter.

With gratitude,
Håvard Horgen Thunold
Michael A. Riegler
Anis Yazidi
Hugo L. Hammer

Reviewer 2 Report

Comments and Suggestions for Authors

The article concerns the use of artificial intelligence (deep learning) for the diagnosis of gastrointestinal diseases. The topic is current and presented in an interesting way. Detailed comments below: 1. Abstract – contains information about the research method, the purpose of the research and the results. It also contains a short background, but in my opinion it is insufficient and should be expanded. 2. Section 1. Introduction – the introduction is written mostly correctly. However, I would suggest emphasizing the purpose of the work more clearly and indicating the research gap. I also have the impression that there is a lack of emphasis on new things and a greater focus not only on artificial intelligence but also on the diagnostics itself. Please consider rebuilding the section. 3. Section 4.2. Clustering Performance Evaluation, Figure 1. presents, in my opinion, a very important scheme of the conducted research. Unfortunately Fig.1. is illegible. I believe that the individual steps of the study should be more exposed. 4. Section 6. Experiments and Results accurately presented and including the necessary elements. 5. Section 7. Discussion is the closing section of the article, but in my opinion, this section should include a discussion of the results obtained by the authors with those of other researchers and present the limitations of the proposed method. I also propose a separate section, Conclusions. The Conclusions section should respond to the purpose of the research and indicate future work.

Author Response

(The authors gave the same response as above.)

Reviewer 3 Report

Comments and Suggestions for Authors
  1. Refine the topic: The topic needs to be applied to a specific image areas rather than being all-encompassing. This network cannot be universally applied.

  2. Structural Improvements: Sections 3 and 4 should be integrated into the related work. Certain methods from section 4 should be incorporated into Section 5 (which should be renamed to section 3).

  3. Methodology Framework Clarification: It is imperative to depict the network architecture systematically throughout the paper. Rewrite the methodology section, beginning with an illustrative framework of the proposed network. Subsequently, elaborate on each aspect of the framework in stages within subsections of the methodology.

  4. Streamlined Experimentation and Results: The Experiments and Results section contains unnecessary narratives. For instance, the explanation in section 6.1 is redundant. Discuss the datasets, then proceed directly to the results. Present both numerical and visual results. Remove any irrelevant content from this section.

  5. Ablation Experimentation: Conduct an ablation experiment and provide a detailed report on the outcomes.

  6. Detailed Image Captions: Enhance image captions for clarity. They should be sufficiently detailed to ensure that everyone can comprehend the results or images presented.

  7. Comparative Analysis: Compare your results with those obtained from other methods in the field.

  8. Inclusion of Evaluation Metrics: Incorporate evaluation metrics for this paper, including but not limited to AUC (Area Under the Curve), and other relevant metrics.

Overall, while the topic of the paper is intriguing, the presentation standard falls below expectations. Given that this is a computer vision paper, it is imperative to provide detailed methodologies and results. Addressing these specific points will elevate the overall quality of the paper.

Author Response

(The authors gave the same response as above.)

Reviewer 4 Report

Comments and Suggestions for Authors

The paper introduced a compelling framework that combines Explainable Artificial Intelligence (XAI) with "explanation-weighted" clustering to gain deeper insights into Deep Learning (DL) models trained on unstructured data. The application of this framework to gastrointestinal tract images, including synthetic pathologies, yields promising results in organizing images based on their pathological diagnoses, with high cluster quality and a Rand index close to one. However, there are important areas for improvement:

1. While the dataset split into training, validation, and test sets is mentioned, it would be advisable to employ K-fold cross-validation for more robust performance evaluation, ensuring that the results generalize well.

2. To provide a more thorough evaluation, the paper should incorporate comprehensive metrics such as ROC curves, AUC, sensitivity, and specificity to assess the model's predictive performance.

3. In addition to cluster quality, it is essential to investigate clustering stability to understand how robust and consistent the clustering results are under different conditions or with variations in the dataset.

4. The authors should consider reviewing recent studies in the field of explainable deep learning, such as "Interpretable graph convolutional network of multi-modality brain imaging for Alzheimer's disease diagnosis" and "Sparse Interpretation of Graph Convolutional Networks for Multi-modal Diagnosis of Alzheimer’s Disease," to ensure their work remains up-to-date and potentially leverage advanced techniques.

Comments on the Quality of English Language

N/A

Author Response

(The authors gave the same response as above.)

Reviewer 5 Report

Comments and Suggestions for Authors

The authors described a method using deep learning to perform a classification task on medical image data. The predictions are further explained with clustering. This is an interesting area of research to show confidence in a prediction model. However, I have major concerns about the methods as outlined below:

- In line 298, you mentioned that the original dataset contains 10,662 labelled instances that are deemed highly accurate. However, only a fraction of these were used in your experiments, requiring augmentation to boost data samples. This is a major flaw that must be addressed by using all the available data samples.

- You created your own pathological markers instead of using existing ones in the dataset. These are very distinctive and extremely likely to influence the classification task which was evident in the 100% predictions achieved. You could have used the real images of polyps rather than creating coloured ellipses on the image. This is a major flaw that must be addressed.  

- You suggested that larger clusters represent more important explanation while smaller clusters represent less important explanation. You need to justify this claim (with real images and corresponding labels). Consider exploring outlier detection as a way to understand the smaller clusters which may indicate higher importance. 

- you claimed in line 260 to 262 that the method will work on any other dataset, without proof. I suggest that you re-run your experiments first with the actual images as indicated in my previous comments. Then, undertake the experiments on another dataset. This will validate your claim.

Other minor to moderate concerns:

- proofread the paper for typo errors and sentence ambiguity. I have highlighted a few lines but this is not exhaustive - lines 25 to 33 contains too many "for example".  line 43 to 44 contains many "based on". line 64 is ambiguous. missing "by" typo error in line 66. 

- Revise the main contributions of the paper. The third and forth bullet points are inaccurate and should be removed especially the last one.

- chapters 2 to 4 could be compressed into a single chapter called "Background and Literature Review". It is too long with mostly generic / descriptive content that could be summarised. These chapters are currently presented in a thesis-like structure and line 265 to 266 confirms this because the authors referred to the paper as a thesis.

Comments on the Quality of English Language

The quality is moderate and can be easily improved. These includes moderate structural issues and typographic errors.

Author Response

(The authors gave the same response as above.)

Round 2

Reviewer 1 Report

Comments and Suggestions for Authors

Authors have addressed all the comment. now it is ready for publication.

Comments on the Quality of English Language

 English very difficult to understand or incomprehensible

Author Response

Thank you for the positive feedback.

Reviewer 3 Report

Comments and Suggestions for Authors

Please accept.

Author Response

Thank you for the positive feedback.

Reviewer 4 Report

Comments and Suggestions for Authors

The authors have addressed my concerns accordingly, so I recommend accepting the manuscript in the current version.

Author Response

Thank you for the positive feedback.

Reviewer 5 Report

Comments and Suggestions for Authors

The authors did not address the main issue in my previous review which is related to the modified dataset used for validation. Specifically, they used "2,056 images, with half being real pathological images, in the form of polyps, and half being 295 pseudo-real, in the form of blue shapes" - see lines 295 - 296. This is inappropriate especially given the critical nature of the task being undertaken. 

Comments on the Quality of English Language

moderate editing of English language required but this does not affect readability.

Author Response

Thank you for your comments. Please find the response letter attached.

Round 3

Reviewer 5 Report

Comments and Suggestions for Authors

My concern is related to the research method where synthetic data was used for validation. This is inappropriate and the paper needs substantial review to address this issue. 

Comments on the Quality of English Language

Moderate correlation required